# Filtered Convolution for Synthetic Aperture Radar Images Ship Detection

**Luyang Zhang** [1,2] , **Haitao Wang** [1,*], **Lingfeng Wang** [3], **Chunhong Pan** [2], **Chunlei Huo** [2], **Qiang Liu** [1] **and Xinyao Wang** [1]

[1] College of Automation Engineering, Nanjing University of Aeronautics & Astronautics, Nanjing 210016, China
[2] National Laboratory of Pattern Recognition, Institute of Automation, Chinese Academy of Sciences, Beijing 100190, China
[3] College of Information Science and Technology, Beijing University of Chemical Technology, Beijing 100029, China
[*] Correspondence: htwang@nuaa.edu.cn; Tel.: +86-189-1382-3432

**Abstract:** Synthetic aperture radar (SAR) image ship detection is currently a research hotspot in the field of national defense science and technology. However, SAR images contain a large amount of coherent speckle noise, which poses significant challenges in the task of ship detection. To address this issue, we propose filter convolution, a novel design that replaces the traditional convolution layer and suppresses coherent speckle noise while extracting features. Specifically, the convolution kernel of the filter convolution comes from the input and is generated by two modules: the kernel-generation module and local weight generation module. The kernel-generation module is a dynamic structure that generates dynamic convolution kernels using input image or feature information. The local weight generation module is based on the statistical characteristics of the input images or features and is used to generate local weights. The introduction of local weights allows the extracted features to contain more local characteristic information, which is conducive to ship detection in SAR images. In addition, we proved that the fusion of the proposed kernel-generation module and the local weight module can suppress coherent speckle noise in the SAR image. The experimental results show the excellent performance of our method on a large-scale SAR ship detection dataset-v1.0 (LS-SSDD-v1.0). It also achieved state-of-the-art performance on a high-resolution SAR image dataset (HRSID), which confirmed its applicability.

**Keywords:** synthetic aperture radar (SAR); remote sensing image ship detection; filter convolution; coherent speckle noise; local weight

## 1. Introduction

Synthetic aperture radar (SAR) is an active microwave detection system that can emit microwaves around the clock and generate high-resolution images using microwaves reflected by objects [1]. SAR can perform large-area detection at night and under adverse conditions and can penetrate vegetation, soil, and lakes, overcoming the limitations of optical and infrared systems. Therefore, SAR images have high research and application value for agricultural surveying and mapping, oceanographic research, ship inspection, and military reconnaissance [2]. However, during SAR image generation, echoes of multiple scattering points are superimposed coherently, which inevitably forms coherent speckles [3,4] (see Figure 1). The existence of coherent speckles reduces the contrast of SAR images, weakens edge details, and significantly reduces the efficiency of SAR image interpretation, retrieval, and other applications, including SAR image segmentation [5], object detection [6], recognition, and classification [7]. In particular, the above-mentioned situation is further exacerbated in the inshore scene, since metallic objects have similar scattering

properties to shores. Therefore, coherent speckling is the primary problem to be solved for understanding and analyzing SAR images.

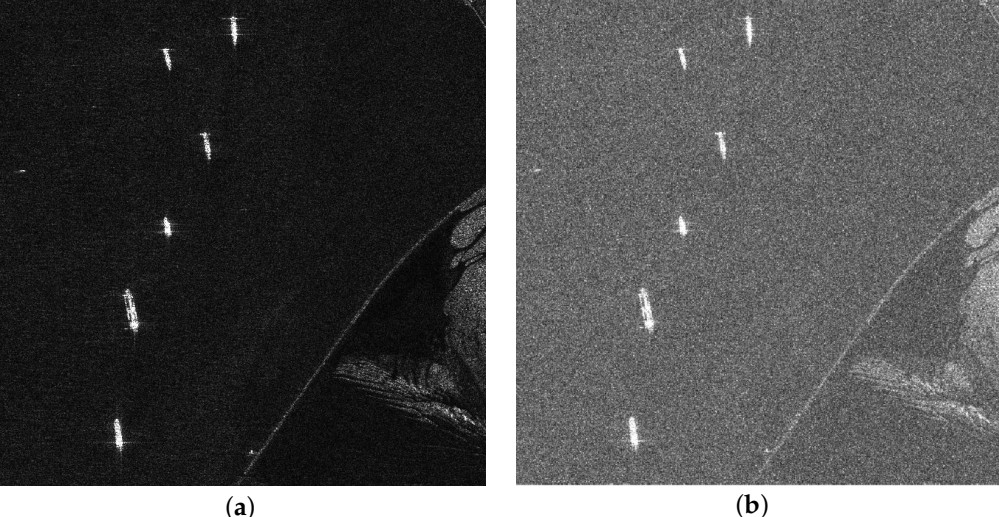

**Figure 1.** Comparison of a clean image and an image contaminated by coherent speckles: (**a**) clean image; (**b**) coherent speckle image (Coherent speckle noise is artificial and has a Gamma distribution.)

Based on the generation mechanism of SAR images, coherent speckle suppression methods have been developed with the advent of SAR images. Researchers first proposed an estimation domain-based method based on natural optical image processing methods, and many studies have confirmed its effectiveness [8,9]. However, statistical models of noise in SAR images are usually built on the premise that the area is homogeneous, and ideal performance can be achieved in homogeneous areas, while in texture-rich areas, the image structure information (edges, textures, and point objects) will be blurred or lost. In addition to the estimated domain-based methods described above, other effective speckle suppression methods exist [10–12]. However, similar to statistical domain-based methods, they often fail to preserve sharp features such as edges and often contain blocking artifacts in denoised images.

In recent years, with the outstanding performance of deep learning in the field of computer vision, researchers have begun to explore methods for speckle suppression based on convolutional neural networks (CNN). Efficient methods have also been proposed, such as DnCNN [13] and ID-CNN [14]. However, existing CNN-based methods usually use the feature mapping model to remove coherent speckle by constructing mapping between the coherent speckle and clean SAR image. This feature mapping model is significantly affected by human factors, which significantly limits the generalization ability of the model. In addition, these methods are derived from general image denoising techniques and are not specialized in speckle noise, and the additional training model makes it impossible to achieve end-to-end training.

To solve these problems, this paper proposes a filtered convolution for SAR image ship detection. Inspired by dynamic convolution [15], filtered convolution can replace classic convolutional layers with plug-and-play and suppress speckle noise while extracting features. The filtering convolutional layer is composed of two important modules: the kernel-generation module and the local weight generation module. The kernel-generation module was implemented using a dynamic convolution structure. We referred to the SENet [16] structure and generated dynamic convolution kernels based on the global statistical properties of the input. The local weight generation module is used to generate local weights based on the local statistical features of the input such that the extracted features contain more local information. In addition, we theoretically and experimentally confirmed that the coupling of the kernel-generation module and local weight generation module can

effectively suppress speckle noise during the feature extraction process. Finally, the proposed filtered convolution was applied to SAR image ship detection, and its effectiveness was verified on challenging datasets LS-SSDD-v1.0 and HRSID.

In short, the contributions of this study are as follows:

- We propose a filtered convolutional layer based on a dynamic convolutional structure. It is composed of a kernel-generation module and a local weight generation module and can replace the traditional convolutional layer with plug-and-play.
- We theoretically confirm that the proposed filter convolutional layer can effectively suppress speckle noise in the feature extraction process and design experiments to verify its effectiveness.
- The proposed filtered convolution is applied to the ship detection task and improves the performance of the baseline method Cascade RCNN for ship detection in SAR images. The experimental results show that our method achieves outstanding performance on LS-SSDD-v1.0 and HRSID.

The remainder of this paper is organized as follows. Section 2 summarizes the related work on ship detection, coherent speckle suppression, and dynamic convolution. Section 3 describes material and methods, and a detailed analysis of its characteristics. Section 4 reports the details of the experiment and results, including the datasets, ablation experiments, and the overall evaluation. Finally, Section 5 presents the conclusions of this study.

## 2. Related Work

### 2.1. Coherence Speckle Suppression Method

With the emergence of a new generation of spaceborne SAR systems, researchers have attracted considerable attention for SAR image processing applications, and research on speckle suppression has been the most extensive. The speckle suppression method is usually designed by combining different estimation domains, estimation criteria, linear minimum mean square error, maximum posterior probability estimation, and probability density estimation models. Examples include the Lee filter [8], Refined-Lee filter [9], Kuan filter [17], Frost filter [18], and Gamma-Map filter [19]. The above method was confirmed from theory and experiments, indicating that the filter derived from the statistical model of coherent speckle can effectively suppress coherent speckle. However, statistical modeling of noise in SAR images is based on the assumption of homogeneous regions. Therefore, the above filters can achieve ideal performance in homogeneous regions, whereas in texture-rich regions, it will lead to the blurring or filtering out of image structure information (edge, texture, and point target). In addition to the estimated domain-based methods described above, some effective speckle suppression methods exist, including wavelet-based methods [10,20,21], block-matching 3D (BM3D) algorithms [11], and total variation (TV) methods [12]. However, similar to statistical domain-based methods, they often fail to preserve sharp features such as edges and often contain blocking artifacts in denoised images.

With the excellent performance of convolutional neural networks in the field of image processing, some researchers have begun to explore CNN-based coherent speckle-suppression methods. Zhang et al [13] experimentally confirmed that residual learning and batch normalization can speed up the training process and improve denoising performance, and they designed a denoising convolutional neural network (DnCNN) to predict the difference between noisy and potentially clean images. The experimental results confirmed that the proposed DnCNN can effectively remove various types of image noise. In [22], a dilated residual network was proposed to denoise SAR images, and a noise suppression model was constructed by learning nonlinear end-to-end mapping between noisy and clean SAR images. It shows excellent performance in quantitative and visual evaluations, particularly in suppressing strong speckle noise. In [14], an image-despeckling convolutional neural network (ID-CNN) was proposed to automatically remove speckling from noisy input images. ID-CNN reconstructs the residual layer of the network to estimate speckling and uses a combination of Euclidean loss and total variation (TV) loss for training,

which significantly improves the performance of speckle suppression. The authors of [23] proposed a down-to-earth deep learning approach to manage despeckled SAR images without using the ground truth (despeckled images). Ref. [24] proposed a multi-scale residual dense dual attention network (MRDDANet) for SAR image denoising. This can effectively suppress speckles and completely preserve the texture details of the image.

### 2.2. Dynamic Convolution

Dynamic filters are dynamically modified or predicted based on input features, which can increase the size and capacity of the model and maintain an efficient inference. In 2016, [25] first proposed a dynamic convolution framework in which filters are dynamically generated according to the input conditions, making this framework more flexible without excessively increasing the number of model parameters. In recent years, dynamic convolutional structures have continuously improved.Moreover, [26] proposed a GaterNet structure for generating dynamic filters, which further improved the performance of dynamic convolutional architectures and had better generalization ability, and [15] proposed an attention-based dynamic convolution structure. Dynamic convolution is dynamically aggregated into multiple parallel convolution kernels according to attention, and attention is related to the input. Increasing model complexity without increasing network depth or width makes the extracted features more representational. Inspired by the progress of attention, [27] separated depth dynamic filters into spatial and channel dynamic filters and proposed decoupling dynamic filters (DDF). DDFs significantly reduce the number of parameters and limit the computational cost to the same level as depthwise convolutions.

### 2.3. SAR Image Ship Detection

In recent decades, ship detection in SAR images has received extensive attention and research from relevant researchers and institutions, and new detection methods have emerged. Most traditional methods are based on the statistical properties of SAR images, including global threshold-based, constant false alarm ratio (CFAR)-based [28], generalized likelihood ratio test (GLRT)-based, transform domain-based, visual saliency-based, and auxiliary feature-based methods . These methods obtain moderate results in specific contexts but require certain priors, are computationally complex, and have poor generalization ability, which cannot meet engineering requirements.

With the wide application of deep learning in the field of computer vision, SAR image ship detection methods based on convolutional neural networks have become mainstream. Ref. [29] designed a Dense Attention Pyramid Network (DAPN) that introduced a convolutional block attention module for adaptive multi-scale SAR image detection. Fu et al. [30] fused attention-guided balanced pyramids and refined heads to detect SAR images using an anchor-free approach and explored a reasonable balance between speed and accuracy. Zhang et al. [31] proposed a ship detector based on multitask learning (MTL-Det) to distinguish ships in SAR images. Modeling the ship detection problem as three collaborative tasks improves the learning of ship-specific features without the additional cost of manual labeling. Xu et al. [32] proposed an optimization method for dynamically learning hyperparameter configurations, which further improved the performance of the SAR image ship detection algorithm by dynamically learning the hyperparameter configuration using deep reinforcement learning (DRL). In [33], the feature extraction and classification methods of different soft computing techniques for land use and land cover were summarized. Soft-computing techniques have been introduced to identify various regions with individual textures and shapes.

## 3. Material and Methods

In this section, we describe in detail the filtered convolutional neural network (F-CNNs). The purpose was to suppress the coherent speckle noise of the SAR image during the feature extraction process and improve the performance of the ship detection model.

Compared to existing speckle suppression methods, the proposed filtered convolution method has the following advantages:

- Filtered convolutional layers are able to suppress speckle noise in the feature extraction process without an additional separate step.
- The extracted features contain more local information, which is beneficial for SAR image ship detection.
- The convolution kernel parameters were learnable and can be updated during the back propagation of the network.

*3.1. Filtered Convolution Kernel Generation*

Inspired by the traditional coherent speckle noise suppression algorithm, we propose a filtered convolution based on the statistical characteristics of an input image or feature map. In filtered convolution, the convolution kernel is generated by the kernel-generation module and local weight generation module. The convolution kernel-generation branch is used to generate the dynamic convolution kernel, and the local weight generation branch makes the network pay more attention to the local area, both of which originate from the input image or feature map.

3.1.1. Kernel Generation

The kernel-generation module is primarily used to generate dynamic convolution kernels according to the global characteristics of the input, and the network structure is shown in Figure 2. We refer to the basic structure of SENet [16]. For the input image or feature map $I \in R^{c \times w \times h}$ (where $h$, $w$, and $c$ are the height, width, and number of channels of input $I$, respectively), we first pass a global average pooling layer to obtain global features. Then, the FC layer is used to transform the number of channels, the ReLU layer is used for non-linearization, and the Norm layer is used for normalization. Finally, the elements of each channel dimension were resized as kernels with $k \times k$.

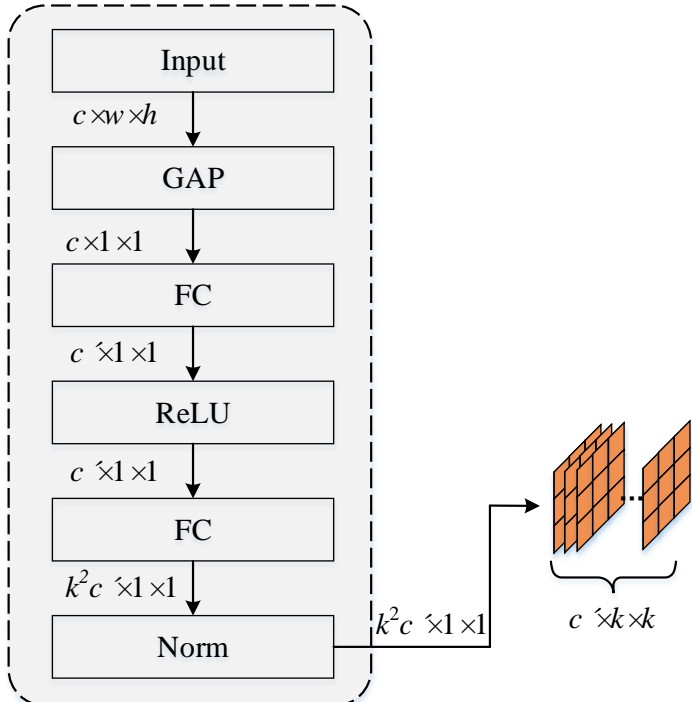

**Figure 2.** The kernel generates the module network structure, where GAP, FC, ReLU, and Norm denote the global average pooling layer, the fully connected layer, the activation function layer, and the normalization layer, respectively.

### 3.1.2. Local Weight Generation

The local weight generation module is mainly used to improve the network's ability to represent local information, which is crucial for both coherent speckle suppression and ship detection tasks. For the input image or feature map $I \in R^{c \times w \times h}$, we first obtain the local statistical properties of each element through a specific multilayer sliding window. The size of the sliding window is $k \times k$ with a stride of 1. Multilayer processing is used to expand the regional receptive field. Then, the $1 \times 1$ convolutional layer was used to resize the feature channel to $k^2$ and normalize it. Finally, the elements at the same position in each channel were resized to $k \times k$ convolution kernels. The detailed structure of the local weight-generation network is shown in Figure 3.

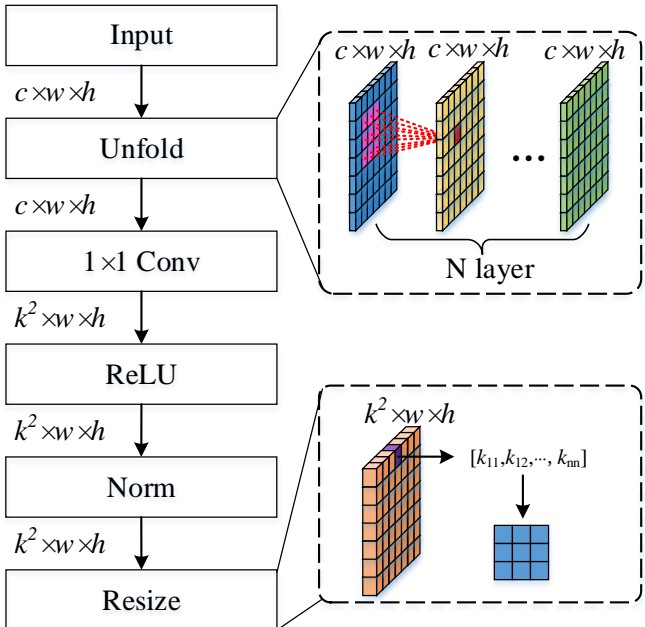

**Figure 3.** Local weight generation module network structure, where Unfold represents the sliding window implementation, and Resize represents the resize element of each channel.

### 3.2. Filter Convolutional Layer

First, we briefly formulate a standard convolution, given an input $I \in R^{c \times w \times h}$, the standard convolution operation at the $i^{th}$ pixel can be written as a linear combination:

$$Y_{(.,i)} = \sum_{j \in \Omega(i)} W[p_i - p_j] I_{(.,j)} + b \tag{1}$$

where $W[p_i - p_j] \in R^{c' \times c}$ is the filter at the position offset between the $i^{th}$ and $j^{th}$ pixels and can also be denoted as an adjacent element. $I_{(.,j)}$ denotes the input feature vector of the $j^{th}$ pixel, and $b \in R^c$ denotes the bias vector. $Y$ denotes the output feature map, and $Y_{(.,i)}$ denotes the output feature vector of the $i^{th}$ pixel. $\Omega(i)$ denotes the convolution window around the $i^{th}$ pixel. In the standard convolution, each convolution kernel is shared among all pixels in the input feature map.

In the proposed filter convolution, the convolution kernel is generated from the input image (feature map), and the detailed network structure is shown in Figure 4. The key technology is to construct the statistical features of the convolution kernel based on the input, in which the kernel-generation module is used to obtain the global statistical features and the local weight module is used to obtain the local statistical features:

$$Y_{(.,i)} = \sum_{j \in \Omega(i)} K_i^g[p_i - p_j] K_c^l[p_i - p_j] I_{(c,j)} \tag{2}$$

where $Y_{(.,i)}$ denotes the output feature value at the $i^{th}$ pixel and $c^{th}$ channel, and $I_{(c,j)} \in R$ denotes the input feature values in the $j^{th}$ pixel and $c^{th}$ channel. $K_i^g \in R^{n \times k \times k}$ is the dynamic convolution kernel, where $K_i^g \in R^{k \times k}$ denotes the filter at the $i^{th}$ pixel. $K^l$ denotes the local weights, with $K_c^l$ denoting the weights at the $c^{th}$ channel.

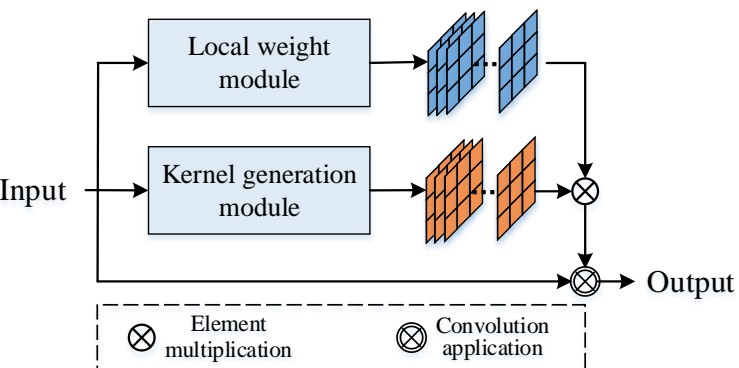

**Figure 4.** Filter Convolutional Layer structure.

*3.3. Suppression of Coherent Speckle Noise*

Several studies have confirmed that speckle noise can be regarded as a multiplicative noise model [34,35]. An input image containing coherent speckles can be expressed as:

$$I = X \times U \tag{3}$$

where $I$, $X$, and $U$ denote the input image, the ideal image without noise, and coherent speckle noise, respectively. Specifically, $X$ and $U$ are independent of one another. To facilitate calculation, it was transformed into a linear model using the Taylor formula:

$$I = \overline{U}X + \overline{X}(U - \overline{U}) \tag{4}$$

where $\overline{U}$ and $\overline{X}$ denote the expectations of $U$ and $X$, respectively.

Based on the above assumptions, the proposed filtering convolutional layer can be formulated as:

$$
\begin{aligned}
S &= (I * K)_{(i,j)} \\
&= \sum_{m,n} I_{(i+m,j+n)} K_{(m,n)} \\
&= \sum_{m,n} (X \times U)_{(i+m,j+n)} K_{(m,n)}
\end{aligned} \tag{5}
$$

where $S$ denotes the element output of the filtered convolutional layer, and $X$ and $U$ are the expected values of the clean image elements and noise components, respectively.

In the filter convolutional layer, $K$ is the multiplication of the output of the kernel-generation module and the local weight generation module and is also regarded as the multiplication of the global and local statistical features ($K^g$ and $K^l$) of the input. Therefore, the convolution process can be expressed as:

$$
\begin{aligned}
S &= \sum_{m,n} (X \times U)_{(i+m,j+n)} K_{(m,n)} \\
&= \sum_{m,n} [\overline{U}X + \overline{X}(U - \overline{U})]_{(i+m,j+n)} [K^g K^l] \\
&= \sum_{m,n} [\overline{U}XK^g K^l + \overline{X}UK^g K^l - \overline{X}\,\overline{U}K^g K^l]_{(i+m,j+n)}
\end{aligned} \tag{6}
$$

The speckle noise $U$ in the SAR image obeys the *Gamma* distribution [36], and the expected value $\overline{U}$ is 1. The probability density function of $U$ can be formulated as:

$$P(U) = \frac{1}{\Gamma(L)} L^L F^{L-1} e^{-LF} \tag{7}$$

where $\Gamma(.)$ denotes the gamma function, and $F \geq 0$ and $L \geq 1$.

Therefore, the first term $\overline{U} X K^g K^l$ in Equation (6) can be rewritten as

$$\overline{U} X K^g K^l = X K^g K^l \tag{8}$$

In a clean SAR image $X$, the global mean characteristic $\overline{X}$ is an extremely small value because most of the background elements are 0. The second term $\overline{X} U K^g K^l$ in Equation (6) is discarded.

In addition, based on the network structure of the global feature module, the elements of $K^g$ can be regarded as the expectation $\overline{XU}$ of the input $I$. Therefore, the third term in Equation (6) can be converted into:

$$
\begin{aligned}
\overline{XU} K^g K^l &= \overline{XU} [\overline{XU}_{(m,n)}] K^l \\
&= [\overline{XU}^2_{(m,n)}] K^l \\
&= (K^2)^g K^l
\end{aligned}
\tag{9}
$$

To sum up the above, the proposed filtered convolution can be expressed as:

$$S \approx \sum_{m,n} [X K^g K^l - (K^2)^g K^l]_{(i+m,j+n)} \tag{10}$$

As shown in Equation (10), the proposed filter convolution has a low correlation with speckle noise $U$ in the feature extraction process. Therefore, we believe that the filtered convolution can effectively suppress the influence of speckle noise on visual tasks, which confirms our idea.

*3.4. Backward Propagation*

The back-propagation of filtered convolutions is essentially the same as that of the standard convolutional layers. We introduce $X_i$ as the $i_{th}$ input feature map and $K_{ij}$ as the input convolution kernel and let $Y_j$ be the $j_{th}$ output feature map. In the backpropagation process, the filter convolutional layer calculates the gradient of the loss function $l$ relative to $X_i$, similar to the previous one, and the gradient of the loss function $L$ relative to $X_i$ is:

$$\frac{\partial L}{\partial X_i} = \sum_j (\frac{\partial L}{\partial Y_j}) * (K_{ij}) \tag{11}$$

where $*$ denotes zero-padding convolution.

The gradient of the loss function $L$ with respect to $K_{ij}$ is:

$$\frac{\partial L}{\partial K_{ij}} = (\frac{\partial L}{\partial Y_j}) * (X_i)^T \tag{12}$$

where $(X_i)^T$ is the transpose of $X_i$.

Compared with traditional convolutional layers, $K_{ij}$ is not a network parameter, but a function of the input $X$. Therefore, the value of the gradient $\frac{\partial L}{\partial K_{ij}}$ is passed to the convolutional layer, and $K_{ij}$ is calculated as a part of the backpropagation algorithm.

*3.5. Filtered Convolution for Ship Detection*

In this section, we will introduce an ship detection algorithm based on filtered convolution. The baseline method adopts the Cascade RCNN method, and the network structure is shown in Figure 5. For the input image, the multi-scale feature map is firstly acquired through the backbone network and FPN; then, the proposal is generated through the cascaded region proposal network; and finally, the bounding box and category of the object are output through the head structure. Specifically, The proposed filtered convolution is applied in the backbone network, replacing the traditional convolutional layers in the first five convolutional layers (based on the analysis in Section 4.6).

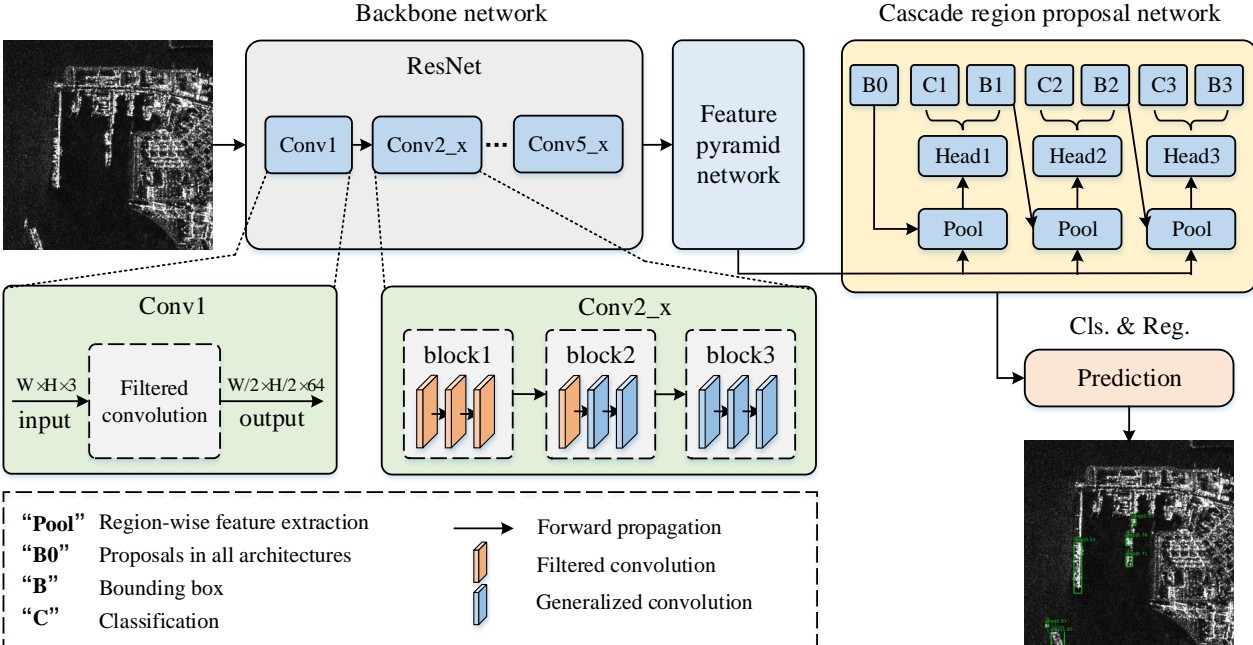

**Figure 5.** Filtered convolution for ship detection algorithm. Cascade RCNN is used as the baseline method and filtered convolution is applied to the backbone network.

## 4. Experiments and Results

In this section, we first introduce two challenging datasets: LS-SSDD-v1.0 [37] and HRSID [38]. The evaluation benchmarks and implementation details of the algorithm are also described. Finally, our method was evaluated on two challenging datasets, and the effectiveness of the proposed filtered convolution was confirmed through ablation studies.

*4.1. Dataset Description*

4.1.1. LS-SSDD-v1.0

The large-scale SAR ship detection dataset-v1.0 (LS-SSDD-v1.0) [37] was derived from Sentinel-1 images in the interferometric wide-format mode of 15 scenes, including ports, straits, and river areas. The size of the original image was approximately $26,000 \times 16,000$ on average, and the large-scale image was split into sub-images of size $800 \times 800$. Finally, 9000 sub-images were generated, in which the sub-images of the first to tenth scenes were used as the training set and the sub-images of the 11th to 15th scenes were used as the test set. The label format of the dataset was PASCAL VOC [39], and the instance category was a ship.

4.1.2. HRSID

The high-resolution SAR image dataset (HRSID) was acquired from Sentinel-1 and TerraSAR-X and used for segmentation and detection tasks [38]. The dataset contained 5604 SAR images and 16,951 instances of HH, HV, and VV polarizations. The native

resolutions were 0.5 m, 1 m, and 3 m, and the image size was $800 \times 800$ pixels. Using Google Earth, the ships in the SAR images were annotated, and pure background samples were discarded. Of these images, 65% were randomly selected as the training set, and the remaining 35% were used as the test set. The label format of HRSID is Microsoft Common Objects in Context (MS COCO) [40], and the instance category was a ship.

### 4.2. Evaluation Protocol

The evaluation benchmarks of LS-SSDD-v1.0 and HRSID adopted the PASCAL VOC and MS COCO benchmarks, respectively. In the experiment, average precision (AP) was used as the main evaluation index and was defined by a precision–recall (PR) curve:

$$\text{Recall} \quad = \quad \frac{TP}{TP + FN} \tag{13}$$

$$\text{Precision} \quad = \quad \frac{TP}{TP + FP} \tag{14}$$

$$\text{AP} \quad = \quad \int_0^1 P(r)dr \tag{15}$$

where $TP$, $FP$, and $FN$ denote the detection results, that is, true positives, false positives, and false negatives, respectively. $P$ denotes the prediction accuracy, and $r$ is the recall. Recall denotes the proportion of the useful part of the detection result to the useful part of the entire dataset, and precision denotes the proportion of the useful part to the entire detection result that is useful.

### 4.3. Implementation Details

Our experiment was carried out on a server running Ubuntu 14.04, Titan X Pascal, and 12G memory.

The proposed method is implemented based on the MMDetection benchmark (MMDetection is an open-source deep learning object detection framework. https://github.com/open-mmlab/mmdetection (accessed on 10 December 2021)) [41]. We used Cascade R-CNN [42] as the baseline method and ResNet50 and ResNet101 as the backbone networks for the experiments. For fairness, all comparative experiments used the same backbone network, and the batch size was set to four because of the limitation of GPU memory. In all the experiments, we used the momentum SGD optimizer to optimize the network, and the momentum and weight decay were 0.9 and $1 \times 10^{-4}$, respectively. The initial learning rate was $5 \times 10^{-3}$. Each training epoch, the learning rate decayed to 0.1 times the original, and the size of the epoch depended on the number of training samples.

### 4.4. Main Result in LS-SSDD-v1.0

We first experimented with our method on the LS-SSDD-v1.0 dataset and compared it with state-of-the-art methods. The comparison methods included single-stage, two-stage, anchor-free, and DETR-based methods. The experimental results in Table 1 show that our method achieves the best performance and improves it by more than 2% compared to the baseline methods. Specifically, an improvement of 1.9% and 2.2% was obtained with ResNet50 and ResNet101 as the backbone, respectively. Two-stage methods have better performance than several other types of methods, especially DETR-based methods, one possible reason DETR-based methods are not suitable for SAR images is due to the large similarity between elements. In particular, the larger-backbone networks showed better improvements. This confirms that our method can effectively improve ship detection performance in SAR images.

We also compared the performance of offshore and inshore scenes. As shown in Table 1, inshore scenarios typically exhibit poor performances. This is understandable because there are many disturbances in the inshore scenarios. However, our method achieves a better boost than the offshore scenario owing to our local weight module. Specifically, the

average prediction (AP) is improved by 1.2% and 2.4% with ResNet50 and ResNet101 as the backbone.

Figure 6 visualizes the detection results on the LS-SSDD-v1.0 dataset. We show a large-scale test image with an image size of 24,000 × 16,000, which is generated by stitching 600 sub-images of 800 × 800.

**Table 1.** Comparison results on LS-SSDD-v1.0.

| Method | Backbone | AP | | |
|---|---|---|---|---|
| | | **Offshore** | **Inshore** | **All** |
| SSD-300 | VGG-16 | 47.7 | 9.0 | 35.4 |
| SSD-512 | VGG-16 | 56.7 | 13.2 | 40.6 |
| YOLOv3 | Darknet-53 | 78.5 | 35.6 | 63.0 |
| RetinaNet | ResNet-101 | 83.7 | 21.0 | 61.9 |
| FCOS [43] | ResNet-101 | 86.5 | 31.0 | 67.5 |
| CenterNet [44] | Hourglass | 87.5 | 29.5 | 68.9 |
| Faster RCNN | ResNet-101 | 87.2 | 35.2 | 68.5 |
| Cascade RCNN * | ResNet-50 | 87.5 | 37.4 | 69.5 |
| Cascade RCNN * | ResNet-101 | 88.0 | 36.8 | 70.8 |
| DETR [45] | ResNet-101 | 61.0 | 20.6 | 44.6 |
| Def-DETR [46] | ResNet-101 | 74.4 | 26.7 | 50.2 |
| Ours | ResNet-50 | 88.6 [+1.1] | 38.6 [+1.2] | 71.4 [+1.9] |
| Ours | ResNet-101 | 89.3 [+1.3] | 39.2 [+2.4] | 73.0 [+2.2] |

* denotes our baseline method. Numbers in [.] indicates the improvement of our method compared to the baseline method.

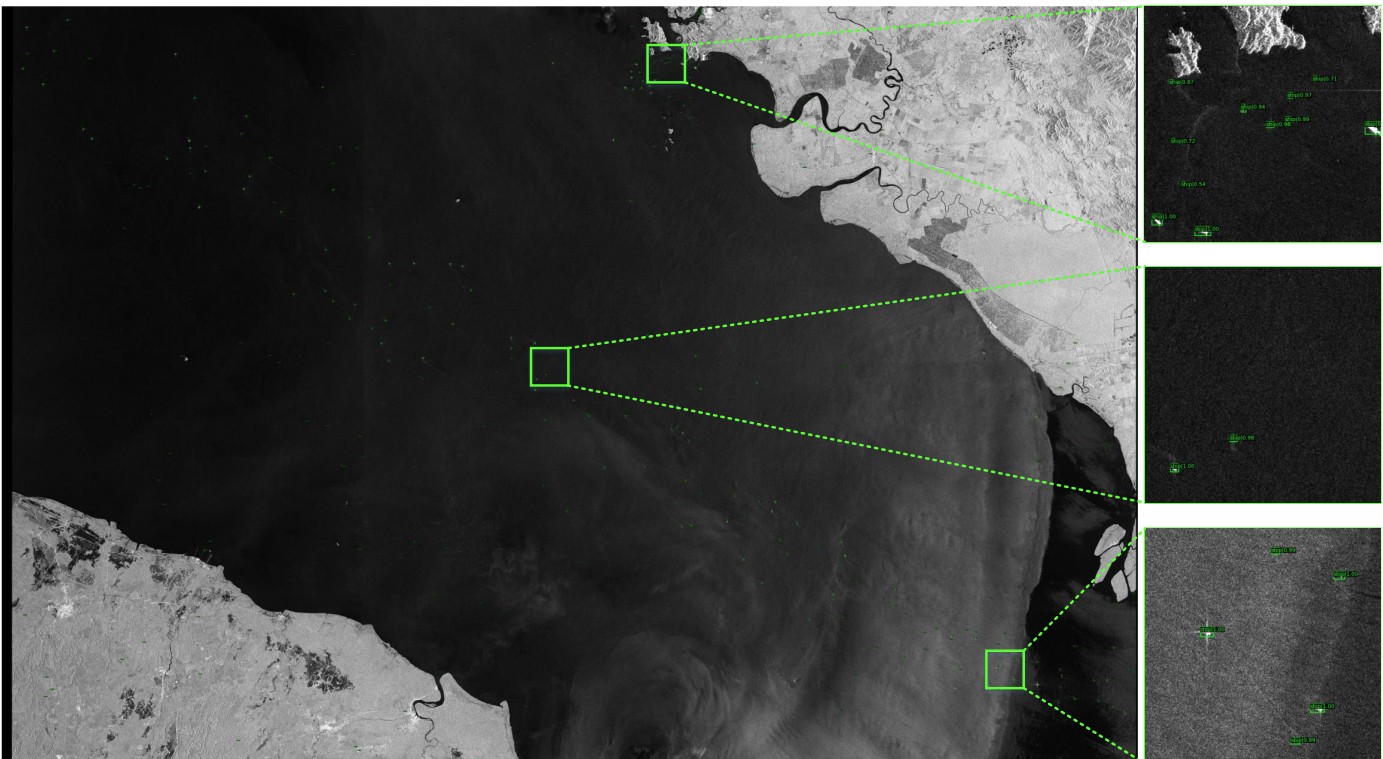

**Figure 6.** The visualization results are on the LS-SSDD-v1.0 dataset, where the text above the bounding box represents the ship category and confidence. The image on the right is a partial enlargement of the large-scale image. Best viewed in zoom in.

### 4.5. Main Result in HRSID

We also experimented with our method on the HRSID dataset and compared it with state-of-the-art methods. In the experiments, Cascade R-CNN was used as the baseline method, and the experimental results are listed in the Table 2. The proposed method obtains 1.4% and 1.8% improvement over baseline methods using ResNet50 and ResNet101 as backbones, respectively.

**Table 2.** Comparison results on HRSID.

| Method | Backbone | AP | $AP_{off}$ | $AP_{in}$ | $AP_{50}$ | $AP_{75}$ | $AP_S$ | $AP_M$ | $AP_L$ |
|---|---|---|---|---|---|---|---|---|---|
| SSD-300 | VGG-16 | 46.6 | 64.1 | 31.9 | 70.0 | 54.2 | 44.9 | 45.6 | 17.2 |
| SSD-512 | VGG-16 | 50.1 | 65.6 | 35.2 | 76.4 | 59.1 | 50.7 | 49.8 | 20.0 |
| YOLOv3 | Darknet-53 | 55.7 | 72.5 | 40.2 | 80.2 | 63.4 | 52.6 | 55.2 | 22.4 |
| Retina-Net | ResNet-101 | 61.3 | 79.6 | 41.3 | 83.7 | 67.7 | 60.1 | 62.7 | 26.5 |
| FCOS [43] | ResNet-101 | 50.2 | 67.3 | 39.1 | 74.2 | 57.6 | 43.9 | 50.5 | 9.4 |
| CenterNet [44] | Hourglass | 53.6 | 70.1 | 39.2 | 78.6 | 64.0 | 52.8 | 57.2 | 19.0 |
| Faster R-CNN | ResNet-101 | 62.5 | 80.7 | 51.4 | 86.9 | 71.6 | 63.3 | 64.4 | 22.0 |
| Mask R-CNN | ResNet-101 | 65.4 | 81.0 | 53.1 | 88.1 | 75.7 | 66.3 | 68.0 | 23.2 |
| Cascade R-CNN * | ResNet-50 | 66.6 | 83.6 | 55.2 | 87.7 | 76.4 | 67.5 | 67.7 | 28.8 |
| Cascade R-CNN * | ResNet-101 | 66.8 | 83.6 | 55.9 | 87.9 | 76.6 | 67.5 | 68.8 | 27.7 |
| DETR [45] | ResNet-101 | 46.8 | 67.2 | 29.5 | 74.5 | 52.9 | 46.5 | 47.6 | 14.9 |
| Def-DETR [46] | ResNet-101 | 51.6 | 70.5 | 40.0 | 78.2 | 54.2 | 50.5 | 53.0 | 23.6 |
| Ours | ResNet-50 | 68.0 [+1.4] | 85.7 [+2.1] | 59.4 [+4.2] | 88.4 | 77.7 | 67.8 | 68.8 | 29.3 |
| Ours | ResNet-101 | 68.6 [+1.8] | 86.2 [+2.6] | 60.8 [+4.7] | 89.2 | 77.6 | 67.4 | 69.2 | 31.1 |

\* denotes our baseline method. Numbers in [.] indicate the improvement of our method compared to the baseline method. $AP_{off}$ and $AP_{in}$ denote the average prediction accuracy for offshore and inshore scenarios, respectively. $AP_{50}$ and $AP_{75}$ indicate the average predictions at confidence thresholds of 0.5 and 0.75. $AP_S$, $AP_M$, and $AP_L$ denote AP for small (area < $32^2$), medium ($32^2$<area<$96^2$), and large (area > $96^2$) objects, respectively.

Furthermore, we compare the proposed method with classical one-stage, two-stage, anchor-free, and DETR-based detectors, and similar results to the previous experiment are obtained. As can be seen from Table 2, the two-stage detector has obvious advantages compared to other types of detectors. This is understandable because the HRSID dataset contains more nearshore samples, making it difficult to distinguish backgrounds and instances and decreasing the performance single-stage and anchor-free detectors. This illustrates that the proposed method can improve the performance of the SAR image ship detectors on different datasets.

The performance of our method in offshore and inshore scenarios is shown in Table 2. The results show that the average precision (AP) improved by 2.6% and 4.7% in offshore and inshore scenarios, respectively. It is worth noting that the inshore scene obtained a greater improvement, which further confirms that the introduction of the local weight module makes the network pay more attention to the local characteristics of the image and improves the performance of the model in the inshore scene of SAR images.

Figure 7 visualizes the detection results on the HRSID dataset. It can be seen that our method achieves outstanding performance in both offshore and inshore scenarios, illustrating the superiority of the method.

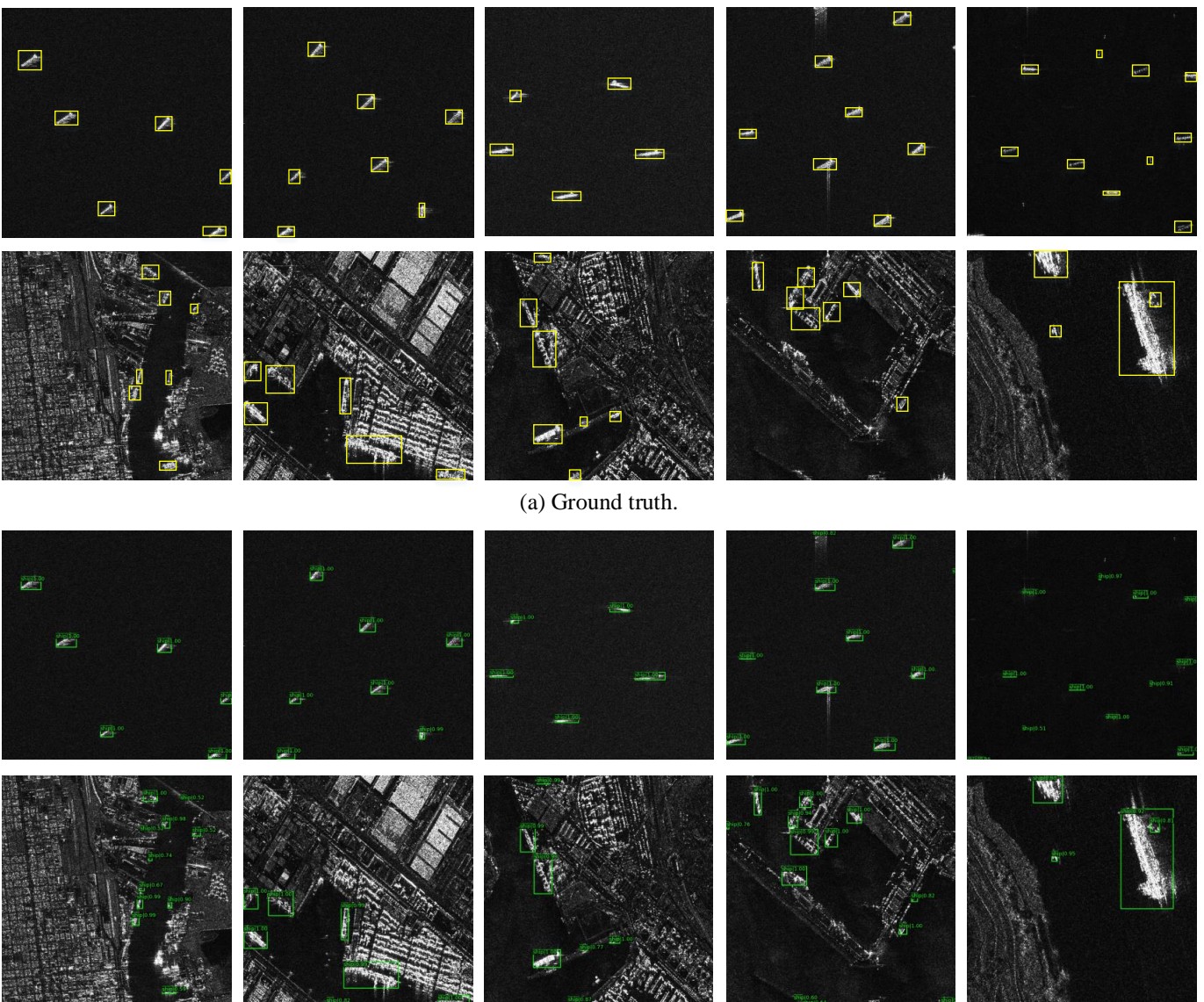

(a) Ground truth.

(b) Predicted results of the proposed method.

**Figure 7.** The visualization results are on the HRSID dataset, where the text above the bounding box represents the object category and confidence.

*4.6. Ablation Study*

In this section, we describe the performance of multiple ablations on the proposed filtered convolution. The default baseline method is Cascade R-CNN, and ResNet50 and ResNet101 are used as backbone networks. All ablation experiments were performed using the HRSID dataset.

4.6.1. Number of Filter Convolutional Layers

In the filtered convolution, the convolution kernel fuses the global and local statistical properties of the input. As the number of network layers increases, the receptive field of the local statistical properties increases. Therefore, filtered convolution cannot be used in all convolutional layers.

Intuitively, filtered convolutions should be applied in the first few layers of the features to effectively suppress the speckle noise from the input image. We experimented with several combinations, and the results are listed in Table 3. C1 represents the first convolutional layer of the backbone, and C2–C7 represent the first two bottlenecks. The experimental

results show that the introduction of filter convolution in the first few convolution layers effectively improves the performance of the model; however, the performance of the model starts to decline at the beginning of C6. This confirms that introducing filtered convolution in the initial stage can suppress speckle noise. However, with the extension of the network, the receptive field of the local weight module is expanded, which limits the improvement in the ship detection task.

**Table 3.** The performance of applying filtered convolutions to different network layers.

| Backbone | C1 | C2 | C3 | C4 | C5 | C6 | C7 | AP |
|---|---|---|---|---|---|---|---|---|
| ResNet-50 | - | - | - | - | - | - | - | 66.6 |
| ResNet-50 | ✓ | - | - | - | - | - | - | 67.4 |
| ResNet-50 | ✓ | ✓ | - | - | - | - | - | 67.4 |
| ResNet-50 | ✓ | ✓ | ✓ | - | - | - | - | 67.3 |
| ResNet-50 | ✓ | ✓ | ✓ | ✓ | - | - | - | 67.5 |
| ResNet-50 | ✓ | ✓ | ✓ | ✓ | ✓ | - | - | **67.7** |
| ResNet-50 | ✓ | ✓ | ✓ | ✓ | ✓ | ✓ | - | 67.2 |
| ResNet-50 | ✓ | ✓ | ✓ | ✓ | ✓ | ✓ | ✓ | 67.0 |

Bold indicates the best performing result.

In addition, [16] confirmed that our kernel-generation module can benefit the feature extraction network. Therefore, this work applies the kernel-generation module alone to the remaining convolutional layers, replacing the traditional convolution. The experimental results are shown in Table 4. Compared to the baseline method, Cascade R-CNN, the introduction of filtered convolution achieved a 1% improvement. In addition, an additional 0.3% improvement was obtained when the kernel-generation module was applied to the remaining convolutional layers. The experimental results confirm the effectiveness of our method.

**Table 4.** Effects of kernel-generation modules.

| Baseline | Group | Loc-Weight | Ker-Generation | | AP |
|---|---|---|---|---|---|
| | | | $Layer_5$ | All | |
| Cascade RCNN | G1 | - | - | - | 66.6 |
| Cascade RCNN | G2 | ✓ | ✓ | - | 67.7 |
| Cascade RCNN | G3 | ✓ | - | ✓ | 68.0 |

$Layer_5$ represents the first five convolutional layers of the backbone network. The local weight generation module is only used in the first five convolutional layers of the backbone network.

In summary, the best performance can be achieved when the filter convolution is introduced in the first five convolutional layers of the backbone, and the kernel generation module is introduced in the subsequent convolutional layers, as detailed in Table 4. Therefore, the G3 combination was also used in the above-mentioned overall evaluation experiments.

4.6.2. Speckle Noise Suppression

To verify the performance of the proposed filtered convolution in suppressing coherent speckles, we extract the first five convolutional layers of the feature extraction network and evaluate their coherent speckle noise properties. For fair experimentation, we also evaluate the resized input images and use the weights from the final trained model. The equivalent number of looks (ENL) and radiometric resolution ($\gamma$) were used to evaluate the speckle noise characteristics of the feature. ENL reflects the smoothness of the image, and $\gamma$ reflects the ability of the SAR system to distinguish the backscattering coefficient of the target. ENL and $\gamma$ can be formulated as:

$$\text{ENL} = \frac{\mu_l^2}{\sigma_l^2} \tag{16}$$

$$\gamma = 10log(\frac{\mu_l}{\sigma_l} + 1) \tag{17}$$

$$= 10log(\frac{1}{\sqrt{\text{ENL}}} + 1)$$

where $\mu_l$ and $\sigma_l$ are the mean and standard deviation of the local region in the input SAR image, respectively.

According to the definition of ENL, a larger value of ENL indicates that the coherent speckle has less influence on the image. The experimental results in Table 5 show that, with the extension of the network layer, the ENL gradually increases, and $\gamma$ also decreases. This illustrates that the convolutional layers can effectively suppress coherent speckles and confirms the effectiveness of the proposed filtered convolution.

**Table 5.** Evaluation of different output feature layers.

| Layer | C0 | C1 | C2 | C3 | C4 | C5 |
|---|---|---|---|---|---|---|
| Output size | 224 | 224 | 112 | 56 | 56 | 56 |
| $\mu$ | 94.69 | 0.38 | 0.39 | 0.39 | 0.41 | 0.43 |
| $\sigma$ | 4833 | 0.07 | 0.07 | 0.08 | 0.07 | 0.07 |
| ENL | 1.55 | 1.95 | 2.14 | 2.11 | 2.30 | 2.51 |
| $\gamma$(dB) | 2.39 | 2.35 | 2.25 | 2.27 | 2.20 | 2.13 |

C0 is the image after the input image is resized, and the image size is 224×224. The larger $\mu$ and $\sigma$ in C0 are due to the normalization of the input image.

We also compared the proposed method with traditional speckle suppression methods, including Lee filtering, Refined Lee filtering, Kuan filtering, Frost filtering, and Gamma-map filtering. For fair experiments, we resized the input image to the feature map size and evaluated it using ENL. The experimental results in Table 6 show that the proposed filtering convolutional layer is overall better than the traditional filter. Although in initial low-level feature layers, the performance of filtered convolution is worse than traditional methods due to the small receptive field of the initial filtered convolution layer, which limits its ability to suppress speckle noise. However, in high-level feature layers, the opposite result is obtained. Therefore, filtered convolutions have better overall performance. Furthermore, it is worth noting that our method is end-to-end without complicated preprocessing procedures.

**Table 6.** The performance of the traditional speckle noise filter, ENL is used as the evaluation index.

| Image Size | Lee | R-Lee | Kuan | Frost | Gamma |
|---|---|---|---|---|---|
| 800 | 1.77 | 1.80 | 1.74 | 1.90 | 2.11 |
| 224 | 1.98 | 2.06 | 1.88 | 1.94 | 2.04 |
| 112 | 1.96 | 2.10 | 1.89 | 2.00 | 2.20 |
| 56 | 2.06 | 2.19 | 1.93 | 2.09 | 2.27 |

4.6.3. Effect of Local Weight Generation Module

To explore the impact of the local weight generation module on the network model, we applied it at different stages of the network. In the experiments, Cascade R-CNN was used as the baseline method, and ResNet50 was used as the backbone network. The difference is that local weight modules are applied to different convolutional layers of the backbone network. For consistency of the experiments, we designed a similar combination to the previous experiments, $Layer_5$ and All, with additional settings for the kernel-generation module. The experimental results are presented in Table 7. Compared to the baseline methods, the performance of the model was significantly improved when the local weight

generation module was introduced. A 0.7% improvement was obtained when introducing the local weight generation module in Layer 5 and a 2.3% improvement in inshore scenes. In particular, when the kernel-generation module is introduced simultaneously, the AP and $AP_{inshore}$ of the model improve by 1.4% and 4.2%, respectively. Figure 8 shows the feature maps of the different methods in the inshore scenarios. Compared with the other methods, our method can better identify some difficult ships in inshore scenes, showing outstanding performance.

The above experimental results confirm our idea that the local weight generation module allows the extracted features to contain more local information, which is beneficial for ship detection in inshore scenes.

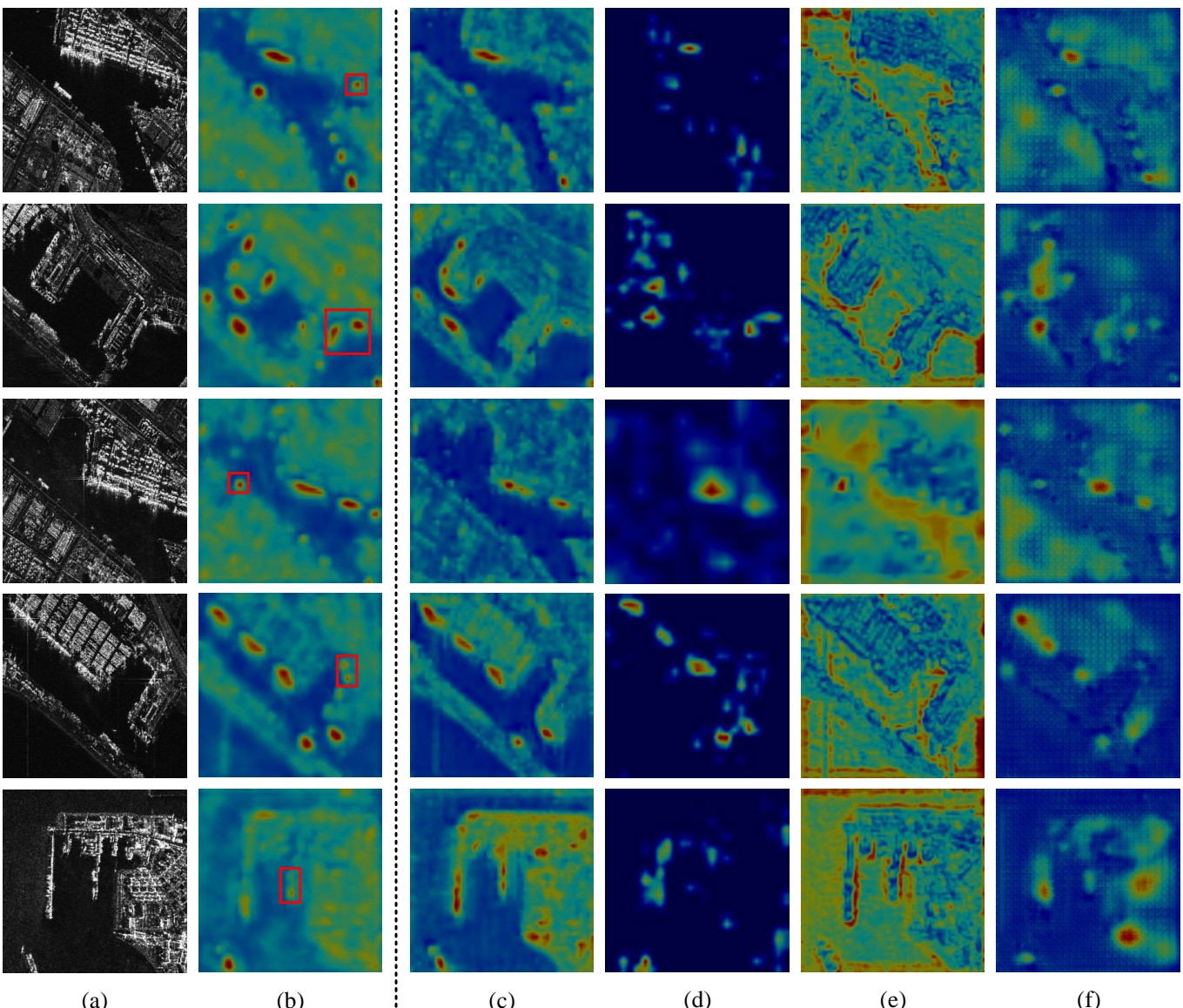

| (a) | (b) | (c) | (d) | (e) | (f) |

**Figure 8.** Feature map visualization results of our method and compared methods. (**a**) Input image. (**b**) Our method. (**c**) Baseline. (**d**) RetinaNet. (**e**) DETR. (**f**) CenterNet. The red box in (**b**) shows the outstanding performance of our method compared to other methods.

**Table 7.** Effects of local weight generation module.

| Method | Ker-Generation | Loc-Weight | | AP | $AP_{inshore}$ |
|---|---|---|---|---|---|
| | | $Layer_5$ | All | | |
| Cascade RCNN | - | - | - | 66.6 | 55.2 |
| Cascade RCNN | - | ✓ | - | 67.3 | 57.5 |
| Cascade RCNN | - | - | ✓ | 66.9 | 56.6 |
| Cascade RCNN * | ✓ | - | - | 67.1 | 56.0 |
| Cascade RCNN * | ✓ | ✓ | | 68.0 | 59.4 |
| Cascade RCNN * | ✓ | - | ✓ | 67.4 | 58.6 |

$Layer_5$ represents the first five convolutional layers of the backbone network. * indicates that the kernel generation module was introduced into the baseline method.

### 4.6.4. Ship Detection With Traditional Filters

To further compare traditional methods, we preprocessed the HRSID dataset using traditional filters and re-evaluated on the baseline methods, the experimental results are shown in Table 8. Specifically, our method achieves the best performance compared to the results after conventional filter preprocessing, especially for the inshore scenes. The baseline method obtains close performance to our method on the dataset after R-Lee and Gamma filter preprocessing, and has better performance in the offshore scenes. This further confirms the better performance of our approach in the inshore scenario. In addition, it is worth noting that our approach is end-to-end without extra preprocessing.

**Table 8.** Comparison with baseline methods (dataset was preprocessed using traditional filters).

| Method | Backbone | Preprocessing | AP | $AP_{off}$ | $AP_{in}$ |
|---|---|---|---|---|---|
| Cascade RCNN | ResNet-50 | Lee | 62.8 | 82.2 | 50.6 |
| Cascade RCNN | ResNet-50 | R-Lee | 67.7 | **86.5** | 54.2 |
| Cascade RCNN | ResNet-50 | Kuan | 58.1 | 80.3 | 47.9 |
| Cascade RCNN | ResNet-50 | Gamma | 66.5 | 85.9 | 56.0 |
| Ours | ResNet-50 | - | **68.0** | 85.7 | **59.4** |

Bold indicates the best performing result.

## 5. Conclusions

This study proposes a filtered convolution structure for ship detection in SAR images. This structure is applied in the backbone network and can replace traditional convolution, which is mainly composed of a kernel-generation module and a local weight generation module. Specifically, the kernel-generation module uses a dynamic convolution structure to generate dynamic convolution kernels based on the global statistical properties of the input. The local weight generation module is based on the local statistical characteristics of the input and is used to improve the network's ability to represent local information. In addition, we theoretically confirmed that the coupling of these two modules can effectively suppress speckle noise in SAR images. Our method was introduced into cascade R-CNN and achieved outstanding performance. Compared with the baseline method on the LS-SSDD-v1.0 dataset, improvements of 1.9% and 2.2% were obtained on the backbone network of ResNet50 and ResNet101, respectively. Improvements of 1.2 % and 2.4 % were also obtained in the inshore scenarios. Similarly, improvements of 1.4 % and 1.8 % were obtained on the backbone network of ResNet50 and ResNet101, respectively, compared to the baseline method on the HRSID dataset. Moreover, 4.2% and 4.7% improvements were also achieved in inshore scenarios.

In this study, we improve the performance of the SAR image ship detection algorithm by constructing a filtered convolution structure. However, filtered convolution can only perform well when applied to the initial layers of the backbone, and specific results are obtained experimentally, which limits its generalization performance.

Therefore, in future research, we tend to try a more generalized structure, similar to the residual structure, which can be adapted to images of any scene. In addition, enhancing the contextual relationship between nearshore instances and the coast is also a direction worth considering.

**Author Contributions:** Conceptualization, L.Z. and L.W.; methodology, L.Z.; validation, H.W., L.W. and C.H.; formal analysis, L.W.; investigation, L.Z.; Resources, H.W. and C.P.; data curation, Q.L. and C.P.; writing—original draft preparation, L.Z.; writing—review and editing, L.Z., C.H. and X.W.; visualization, L.Z. and X.W.; supervision, H.W.; project administration, H.W.; funding acquisition, H.W. and L.W. All authors have read and agreed to the published version of the manuscript.

**Funding:** This work was supported by Nondestructive Detection and Monitoring Technology for High Speed Transportation Facilities, the Key Laboratory of Ministry of Industry and Information Technology, and the Fundamental Research Funds for the Central Universities (NO.NJ2020014), Fund of Fundamental Research Funds for the Central Universities (buctrc202221).

**Data Availability Statement:** The data presented in this study are available on request from the first author.

**Conflicts of Interest:** None of the authors have potential conflict of interest to be disclosed.

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
