# Peer review of "Filtered Convolution for Synthetic Aperture Radar Images Ship Detection"

_remotesensing, doi:10.3390/rs14205257_

Round 1
Reviewer 1 Report
Dear Authors,
my final decision for your paper is to reject it. I personally do not like papers which cover different topics and seems an ensemble of results. In your case, you cover the topics of SAR speckle reduction and object detection proposing your own deep learning technique. This implies a lack of state of the art and proper referencing to previous studies. My suggestion is to divide this paper in two parts: 1) the SAR speckle reduction which you can resubmit in the Special Issue Pattern Recognition in Remote Sensing 2) submit the object detection part (which should actually be titled ship detection as this is the target you demonstrate your approach) in a proper special issue I am sure you could find in the Remote Sensing journal. Of course both papers would need an extensive revision to make each manuscript more complete and specific to the topic addressed. I hope you take my decision as a positive suggestion to better value your work.
Author Response
Response to Reviewer 1 Comments
Comments 1:
My final decision for your paper is to reject it. I personally do not like papers which cover different topics and seems an ensemble of results. In your case, you cover the topics of SAR speckle reduction and object detection proposing your own deep learning technique. This implies a lack of state of the art and proper referencing to previous studies. My suggestion is to divide this paper in two parts: 1) the SAR speckle reduction which you can resubmit in the Special Issue Pattern Recognition in Remote Sensing 2) submit the object detection part (which should actually be titled ship detection as this is the target you demonstrate your approach) in a proper special issue I am sure you could find in the Remote Sensing journal. Of course both papers would need an extensive revision to make each manuscript more complete and specific to the topic addressed. I hope you take my decision as a positive suggestion to better value your work.
Answer by Authors:
Thank you very much for your valuable comments and suggestions! Following your comments, we have re-evaluated our work. We are very sorry to make you understand it as a work that covers different topics due to our inaccurate description. Unlike traditional speckle noise suppression methods, the proposed filtered convolution is not an independent image processing method and cannot be used independently. Filtered convolution is an effective structure in the backbone network of the object detection algorithm, so that it can obtain more effective features, thereby improving the performance of the SAR image ship detection algorithm. Specifically, we detail the application of the proposed filtered convolution in the ship detection algorithm in Section 3.5. Please kindly check Section 3.5 on page 8.
Best regards.
Yours sincerely,
Luyang Zhang

Reviewer 2 Report
Dear Authors,
The article was well written and makes an important contribution to the SAR community regarding the effect of speckle noise and the application of filtering to attenuate its effect on images for target detection.
The results were interesting, but I felt the lack of a statistical test of significance in such a way that it can be tested whether the improvements achieved are significant or not. In addition, I missed an explanation of the time taken to apply the proposed method on the datasets versus the traditional methods.
Also, I have just a few more comments:
1) Figure 1 (p. 2) - What is the source of the SAR images that were used to illustrate the effect of speckle noise? I suggest including the source of the images.
2) Table 1 (p. 10) - What is AP? Describe before or else make it explicit in the Table.
3) Table 2 (page 11) - Describe what each acronym AP50, AP75, APS APM and APL is.
Author Response
Response to Reviewer 2 Comments
Comments 1:
The results were interesting, but I felt the lack of a statistical test of significance in such a way that it can be tested whether the improvements achieved are significant or not. In addition, I missed an explanation of the time taken to apply the proposed method on the datasets versus the traditional methods.
Answer by Authors:
Thank you very much for your valuable comments and suggestions! Following your comments, we have added statistical significance to the experimental results. Please kindly check Table 1 and Table 2 on Pages 11 and 12.
In addition, unlike traditional speckle noise suppression methods, the proposed filtered convolution is not an independent image processing method and cannot be used independently. Filtered convolution is an effective structure in the backbone network of the object detection algorithm, so that it can obtain more effective features, thereby improving the performance of the SAR image ship detection algorithm. Therefore, we did not compare the time taken by the proposed method applied to the dataset compared to traditional methods.
Comments 2:
Figure 1 (p. 2) - What is the source of the SAR images that were used to illustrate the effect of speckle noise? I suggest including the source of the images.
Answer by Authors:
Following your comment, we have added a note to Figure 1 explaining the source of the image and the effect of speckle noise. It is generated by artificial simulation in order to show it visually. Please kindly check Figure 1 on Page 2.
Comments 3:
Table 1 (p. 10) - What is AP? Describe before or else make it explicit in the Table.
Answer by Authors:
Following your comments, we describe AP before Table 1. Please kindly check Section 4.4 on Page 10, second paragraph, line 326.
Comments 4:
Table 2 (page 11) - Describe what each acronym AP50, AP75, APS APM and APL is.
Answer by Authors:
Following your comments, we annotated the definitions of AP50, AP75, APS APM and APL at the bottom of Table 2. Please kindly check Page 12, Table 2.
Best regards.
Yours sincerely,
Luyang Zhang

Reviewer 3 Report
The subject has been studied extensively in the past decades since SEASAT. This manuscript reported object detection in SAR with a method of filtered convolution. The SAR imagery data was single polarization; thus, the objection detection relies on space-radiometric information. The method of filtered convolution is not even a variant of a CNN-based neural network. The primary concerns are as follows.
- The writing is hard to read and follow in that the objective is out of focus. One example is, in the title, it was called for “object,” but indeed, it was only the ship target.
- By filtered convolution, is that to apply a speckle-filtering as a pre-processing of the SAR image?
- This paper applied the Gamma distribution to filter the speckle in SAR data, followed by a CNN-based neural network to train and detect the ships. However, many reports showed the K-distribution sea speckle (clutter). Since K-distribution is a two-parameters model, it is suspected that the “filtered convolution” works well in most sea clutter.
- In conclusion, the authors claimed, "This study proposes a filtered convolution structure for object detection in SAR images. This structure is applied in the backbone network and can replace traditional convolution." The authors should provide more varieties of SAR targets over a heavy clutter background to prove such modification was not objective-oriented (only for SAR ship detection). After all, detecting ships over clam to the moderate rough sea is not difficult, as a rich body of papers has shown in the literatures.
- The ship detection problem has been extensively studied. Based on the limited datasets and results, it is not justified that the paper warrants a scientific publication. The authors perhaps should demonstrate ship identification - a more challenging problem.
Author Response
Response to Reviewer 3 Comments
Comments 1:
The writing is hard to read and follow in that the objective is out of focus. One example is, in the title, it was called for “object,” but indeed, it was only the ship target.
Answer by Authors:
Thank you very much for your valuable comments and suggestions! Following your comments, we have corrected the irregularity by replacing all "object detection" with "ship detection", including the title of the paper.
Comments 2:
By filtered convolution, is that to apply a speckle-filtering as a pre-processing of the SAR image?
Answer by Authors:
Following your comments, we have re-evaluated our work. Different from traditional speckle noise suppression methods, the proposed filter convolution is not a preprocessing of SAR images, and cannot be used independently. Filtered convolution is an effective structure in the backbone network of the object detection algorithm, so that it can obtain more effective features, thereby improving the performance of the SAR image ship detection algorithm. Specifically, we detail the application of the proposed filtered convolution in the ship detection algorithm in Section 3.5. Please kindly check Section 3.5 on page 8.
Comments 3:
This paper applied the Gamma distribution to filter the speckle in SAR data, followed by a CNN-based neural network to train and detect the ships. However, many reports showed the K-distribution sea speckle (clutter). Since K-distribution is a two-parameters model, it is suspected that the “filtered convolution” works well in most sea clutter.
Answer by Authors:
Following your comments, we checked some references on K-distribution sea speckle. As you mentioned in your comment, some papers claim that K-distribution sea speckle is another important influencing factor on the image processing of sea SAR images. This has to be a challenging issue and we would like to explore it in future research.
Comments 4:
In conclusion, the authors claimed, "This study proposes a filtered convolution structure for object detection in SAR images. This structure is applied in the backbone network and can replace traditional convolution." The authors should provide more varieties of SAR targets over a heavy clutter background to prove such modification was not objective-oriented (only for SAR ship detection). After all, detecting ships over clam to the moderate rough sea is not difficult, as a rich body of papers has shown in the literatures.
Answer by Authors:
Thank you very much for your valuable comments and suggestions! Following your comment, we correct the misrepresentation and use ship detection instead of object detection in the title and main text. This is limited by the dataset category.
In adddition, as you mentioned in your comment, identifying more types of objects in SAR imagery is a challenging problem and an urgent need to solve. However, there is a lack of well-annotated SAR image ship detection datasets covering multiple categories, such as aircraft carriers, warships, and cargo ships. We will try to construct a more challenging dataset to help future research.
Comments 5:
The ship detection problem has been extensively studied. Based on the limited datasets and results, it is not justified that the paper warrants a scientific publication. The authors perhaps should demonstrate ship identification - a more challenging problem.
Answer by Authors:
Following your comment, we have re-evaluated our work. As you mentioned in your comment, ship detection has been widely studied in the field of optical remote sensing images and natural images. However, ships in SAR images are still difficult for two main reasons: 1) lack of datasets containing fine-grained categories, 2) difficulty in detecting ships in inshore scenes. We will explore the above challenging problem in future research.
Best regards.
Yours sincerely,
Luyang Zhang

Reviewer 4 Report
General comments:
The manuscript proposes a filter convolution to replace the traditional convolution layer to suppress coherent speckle noise while extracting features for target detection. The filter convolution is generated by two modules of the kernel-generation modual and local weight generation module. Two datasets are used to illustrate the efficiency of the proposed method, but the efficiency and advantage of the method should be illustrated for ships with different features. It is limited by only using the detection accuracies that also has limited improvement.
Specific comments:
1. The accuracies show that the proposed method has better performance over ship detection. It is suggested to illustrate the mechanism for the performance. Generally, ships have strong backscattering on sea are less effected by speckle noise for target detection. Could it improve the detection capability of small ships with weak backscattering? It is suggested to illustrate it.
2. There are similar improvements for the method over the two dataset. There are different resolutions for the two dataset. It is suggested to illustrate the detection mechanism of the method over ships with different features.
3. The optimal layer is selected for the filter convolution by making experiments. The accuracy improvement may be limited if we could not select the optimal layer in prictical applications. Please clarify that.
4. It is suggested to give the ground truth of the ships in the visualization results.
5. It is suggested to compare the results derived from the proposed method using unfiltered data with that of the baseline method using data filtered by traditional filtering method.
6. Page 14, line 384 and 386, ENI should be ENL.
Author Response
Response to Reviewer 4 Comments
Comments 1:
The accuracies show that the proposed method has better performance over ship detection. It is suggested to illustrate the mechanism for the performance. Generally, ships have strong backscattering on sea are less effected by speckle noise for target detection. Could it improve the detection capability of small ships with weak backscattering? It is suggested to illustrate it.
Answer by Authors:
Thank you very much for your valuable comments and suggestions! Following your comment, we describe the mechanism of ship detection based on filtered convolution in Section 3.5. Please kindly check Section 3.5 on page 8.
In addition, as you mentioned in your comment, backscatter is also an important influencing factor for ship detection. The issue of backscatter was not explored in this article due to the lack of corresponding statistics in the available dataset. We will construct an evaluation benchmark and explore the corresponding solution scheme in our future study.
Comments 2:
There are similar improvements for the method over the two dataset. There are different resolutions for the two dataset. It is suggested to illustrate the detection mechanism of the method over ships with different features.
Answer by Authors:
Following your comments, we added a description of the detection mechanism in section 3.5. The SAR image ship detection algorithm based on filtered convolution has strong generalization and can adapt to different scales of input images. Please kindly check Section 3.5 on page 8.
Comments 3:
The optimal layer is selected for the filter convolution by making experiments. The accuracy improvement may be limited if we could not select the optimal layer in prictical applications. Please clarify that.
Answer by Authors:
Thank you very much for your valuable comments and suggestions! Following your comments, we discuss this issue in the conclusion and make it a future research task. Please kindly check the Para 2 of the Conclusion on Page 18.
Comments 4:
It is suggested to give the ground truth of the ships in the visualization results.
Answer by Authors:
Following your comment, we have updated the visualization and the ground truth of the image is given. It is worth noting that we only updated the visualization results in Figure 7, ignoring Figure 6, due to the fact that the visualized bounding box in Figure 6 has a smaller scale and is not easily observed. Please kindly check in Figure 7 on Page 13.
Comments 5:
It is suggested to compare the results derived from the proposed method using unfiltered data with that of the baseline method using data filtered by traditional filtering method.
Answer by Authors:
Following your comment, we experiment with the baseline method after preprocessing using traditional filtering methods and compare the results of the proposed method using unfiltered data. Please kindly check the Section 4.6.4 on Page 16.
Comments 6:
Page 14, line 384 and 386, ENI should be ENL.
Answer by Authors:
Following your comments, we corrected the wrong statement and changed ENI to ENL. Please kindly check Page 14, line 394-396.
Best regards.
Yours sincerely,
Luyang Zhang

Round 2
Reviewer 1 Report
Accepted.
Reviewer 3 Report
The authors have addressed my comments mostly.
Reviewer 4 Report
The reviewer has no further comments. The reviewer recommends a weak acceptance due to the limited performance of the filtered convolution.